

# Public priorities on locally-driven sea level rise planning on the East Coast of the United States

Adam T. Carpenter

Water Policy and Leadership, American Water Works Association, Washington, D.C., United States of America
Environmental Science and Policy, George Mason University, Fairfax, VA, United States of America

## ABSTRACT

Sea level rise poses a substantial concern to communities worldwide. Increased inundation, storm surge, saltwater intrusion, and other impacts create challenges which will require considerable planning to address. Recognizing the broad and differing scope of sea level rise issues and the variability of policy options to address them, local planning frameworks are necessary in addition to tools and resources available from state and federal governments. To help assess priorities and preferences on sea level rise planning, a survey of 503 persons affiliated with coastal communities on the East Coast of the United States was conducted in December 2017. This survey studied key aspects locally-driven sea level rise plans, including planning priorities, funding options, methods to resolve conflict, and potential responses. Six key findings address these and other concerns to provide the foundation of a locally driven framework for public officials.

Corresponding author
Adam T. Carpenter,
acarpen8@masonlive.gmu.edu

# INTRODUCTION

Sea level rise (SLR) poses a serious and ongoing set of challenges to coastal communities globally. With global sea level having already increased by 0.18–0.20 m (0.58–0.67 ft) since 1900, there is a need to plan and prioritize to eliminate impacts where possible and reduce their harm when they cannot be eliminated (*USGCRP, 2018*). Although sea level rise is only one of many impacts of climate change, it has the potential to seriously impact low-lying coastal communities. Several national and multinational bodies project considerable sea level rise by 2100, including the U.S. Global Change Research Program with a 0.30–2.5 m increase relative to the year 2000, and the Intergovernmental Panel on Climate Change (IPCC) with a relative change of 0.26–0.98 m increase in 2081–2100 compared to 1986–2005 (*USGCRP, 2017*; *IPCC, 2013*).

There are several general approaches to decision making around adaptation that could apply to locally-driven planning decisions. The literature surrounding climate change adaptation decision support has expanded considerably over the last two decades (*Palutikof, Street & Gardiner, 2019*). Approaches include, but are not limited to, real options (*Dobes,*

**Table 1**  Summary of selected states with SLR policies (*Carpenter, 2019*).

| State | Key policy |
| --- | --- |
| California | State planning guidance to municipalities |
| Maryland | Strict limits on state involvement in SLR areas |
| North Carolina | Projections limited to 30 years in the future |
| Virginia | Executive order organizing state agencies |

*2008*), robust decision-making (*Groves & Lempert, 2007*), and scenario analysis (*Swart, Raskin & Robinson, 2004*). Each of these approaches uses different processes to approach complex decisions. For example, in a real options approach, adaptation measures can be planned in a manner that allows them to be exercised in the future as scenarios unfold, acknowledging the uncertainty associated with future conditions (*Dobes, 2008*; *Linquiti & Vonortas, 2012*). In a real options approach, some decisions are intentionally preserved for a future date in order to optimize protection while reducing waste, recognizing this uncertainty in today's knowledge as unavoidable until more is known in the future. For example, *Dobes (2008)* describes that planning for a sea wall could begin immediately while the decision of how tall the wall should be could be deferred until more information is known on anticipated future sea levels. Information on public priorities on SLR planning can be utilized in all of these processes, although how specifically will depend on the specific process and the decisions under consideration.

There are a number of recent activities for assessment and planning for sea level rise and coastal flooding in Eastern U.S. states and elsewhere (*Eastern Research Group, 2013*; *Hinkel et al., 2010*; *Miller, Kaminski Leduc & McCarthy, 2012*). Among these are risk-assessments that have taken place in several large US cities, as well as a handful of comprehensive state policies to address various aspects of SLR. Table 1 provides information about several examples, including Maryland and California on the state level (although California is not an Eastern state, it serves as a useful point of reference), and New York City and Miami, Florida on the local level (*California Coastal Commission, 2015*; *Griffin et al., 2008*; *New York Academy of Sciences, 2015*; *Ruvin et al., 2014*). As an example on the opposite end of the policy spectrum, North Carolina has imposed substantial limits on official sea level rise projections and how those projections can be used (*North Carolina General Assembly, 2011*; *Overton et al., 2015*). Other states, such as Virginia, have committed to taking action but are early in developing state-level plans (*Commonwealth of Virginia Office of the Governor, 2018*).

The federal National Flood Insurance Program (NFIP) also assists in reducing the financial risks related to flooding (*Chivers & Flores, 2002*). Within that program, the Community Rating System (CRS) is a mechanism that encourages floodplain management activities in exchange for reduced premiums for NFIP policies for homes and businesses (*FEMA, 2017*). A variety of community-level toolkits and example processes to address sea level rise or broader climate change concerns have also been proposed. For example, a coastal resilience toolkit for New Jersey (USA) provides a complication of municipal maps and mapping tools, vulnerability checklists, an overview of SLR science, and case

studies discussing how these tools have been implemented (*Auermuller & Maxwell-Doyle, 2013*). A NOAA research report walks through methods of selecting SLR scenarios relevant to a specific community and methods for incorporating those projections into various planning processes (*Marcy et al., 2012*). The complexities of adapting to SLR, including challenges around uncertainties that may not be clarified until far into the future and multiple barriers to implementing adaptive measures, may necessitate different strategies than traditional risk management techniques (*May & Plummer, 2011*; *Moser, Ekstrom & Kasperson, 2010*). For example, *Martin et al. (2011)* discusses structured decision making, where larger management issues are broken into smaller components where options are thoroughly analyzed and acted upon, as a methodology to make decisions around sea level rise. Practices such as building codes, elevation standards, and insurance are also pieces of the planning puzzle, but do not by themselves represent comprehensive planning (*Eastern Research Group, 2013*).

Not fully addressed in these programs and discussions, however, is an understanding of public priorities and preferences to assist in developing locally driven sea level rise plans. This work sought to bridge this gap by examining public perceptions on a number of facets of sea level rise planning and translating that information into key findings in a format accessible to public officials that can be used to help develop local sea level rise plans. Building sea level rise policies around local priorities and preferences is not a guarantee for successful protection, as many technical and economic barriers are likely to exist to implementation (*Moser, Ekstrom & Kasperson, 2010*). Although sea level rise planning inherently addresses only some of the impacts of climate change, it may help to catalyze action on other impacts as well. A locally-driven approach has the benefit of being informed by the community and therefore necessarily having some degree of built-in support, helping to strive towards resilience. This study examined perceptions of public priorities on sea level rise planning (and related factors that may influence planning processes), identifies a series of priorities and preferences of those who live in, work in, or regularly visit coastal communities on the East Coast of the United States, and develops six key findings for consideration by public officials in developing locally-driven sea level rise plans.

## MATERIALS & METHODS

Prior to developing the primary survey utilized in this study, a pilot study was developed to help inform the questions and options. This study, which is described in detail in *Carpenter (2018)*, asked three questions to help better understand the breadth of viewpoints on potential elements of locally developed sea level rise plans, acceptable sources to fund the development of the plan and implementation of its recommendations, and potential methods to resolve conflicts that may arise in the development of a sea level rise plan. In this pilot, respondents were requested to provide at least five potential SLR priorities, five funding mechanisms, and five methods to resolve conflict, all of which helped to inform the larger survey instrument.

Building upon the insights of the pilot study, a more in-depth survey instrument was developed to gain more detailed insights on public perceptions of sea level rise planning

priorities. Between December 20 and December 22, 2017, a survey of was conducted of 503 individuals who live in, work in, and/or regularly visit a coastal community on the East Coast of the United States. The study was reviewed by the George Mason University institutional review board (approval number IRBNet 1168842-1). The execution of the survey was performed by Survata, the survey administrator. Written consent was obtained electronically from all participants, and no personally identifiable information was collected.

The survey administrator, who retains a large (many times the sample size needed for this study) potential survey pool designed to be as representative as feasible, targeted potential respondents through a combination of location (US East Coast states), basic eligibility (aged 18+), and through a series of screening questions to assure that they work in, live in, and/or regularly visit a coastal community on the US East Coast. All potential respondents were also required to first read and agree to the information on the consent form prior to proceeding to the screening questions. Respondents were then asked for their affiliation, if any, with US coastal communities. Only those respondents who self-reported that they worked in, lived in, or regularly visited a coastal community on the US East Coast proceeded to the rest of the survey. Although this sampling methodology was intended to be as representative as possible, and survey invitations were sent out randomly amongst this pool until the targeted sample size was reached, it was not without limitations. Although most demographics well well-represented, female respondents considerably outnumbered male respondents, and the information collected does not by itself explain whether this was because of a difference in interest or some other reason. Much like other survey methodologies, little information is known about those who did not participate.

The 503 respondents who passed the screening question were presented with 14 subject matter questions, plus 10 demographic questions. Respondents were not provided with any additional information beyond what is included in the informed consent form, the study's introduction, and the questions themselves, meaning that the respondent's responses are largely reflective of their pre-existing knowledge (or lack thereof) of the topic. Specific information about individual respondent's risks could not be provided because the availability of risk information varies substantially from one location to another. The information gathered includes topics such as relative priorities of other topics in relation to sea level rise, important components for a local plan, preferred funding methods, conflict resolution options, and other facets of planning. Each individual question is discussed in the results (with the full data set available in the supplementary files), and the exact wording, possible responses, and Likert-type scales are shown in the "questions and response meanings" supplementary file. Demographics examined included income, education, ethnicity, political party affiliation, self-reported level of environmentalism, age group, gender, and location.

Likert-type responses are a method that can measure perceptions that are not directly observable or measurable through other means (*Boone & Boone, 2012*; *Carifio & Perla, 2008*). Likert-type responses are often seen on an integer scale from 1 to another number (often 5, but other scales are also used) (*Boone & Boone, 2012*). A basic example in the medical community is "how much pain are you feeling" which is not currently possible

to directly measure, but a patient can clearly identify their perception along a well-marked scale. In a sea-level rise context, it often is possible to model or project how likely or unlikely various adverse events are today and in the future. However, the *perceptions* of residents on how vulnerable they are now and how vulnerable they may be in the future to adverse events is most directly assessed by inquiring. In order to keep the greatest uniformity to the responses, a Likert-type response assigns both directionality (often 1 being least) as well as meanings to numbered responses (*Carifio & Perla, 2008*). Individual Likert-type data are related to but distinct from Likert scales, which use the accumulation of several Likert-type data points to identify a cumulative assessment of some question. This study uses a considerable number of Likert-type data points but does not use Likert scales.

The statistical analysis of Likert-type responses can be controversial, especially the use of parametric tests, meaning that statistical tests and their interpretation had to be chosen carefully (*Boone & Boone, 2012*; *Carifio & Perla, 2008*). Some sources suggest that Likert-type data are appropriately analyzed using parametric tests so long as the scale is clearly labeled (such that all respondents can identify the directionality and meaning of their response) and the respondent pool is sufficiently large, typically above at least 30 responses (*Carifio & Perla, 2008*; *Sullivan & Artino, 2013*). However, others suggest that Likert-type data may be inappropriate for parametric tests regardless of those qualities (*Boone & Boone, 2012*; *Clason & Dormody, 1994*). Because of this controversy around Likert-type data, relationships in this study were examined using non-parametric tests, although information about many of these relationships using parametric tests is available in *Carpenter (2019)*. The use of means also has some controversy for Likert-type data, and is considered by some to be entirely appropriate for larger sample sizes so long as the scale is clearly marked and only similar data (i.e., from the same question) are compared (*Carifio & Perla, 2008*; *Sullivan & Artino, 2013*). Given the large sample size and clearly marked scale, this analysis takes a hybrid approach by providing descriptive statistics including means, although those means are used to compliment other descriptions of frequencies of responses. Significance of relationships within the data was determined primarily with independent samples Kruskal-Wallis tests (non-parametric), using a significance level of 0.05.

There are hundreds of potential correlations between the responses to each question (and each component within each question) and the demographics collected. For most questions, Kruskal-Wallis Tests (using a significance level of 0.05) were used to identify which demographics likely influenced the responses to each component of the question. This results in a series of results indicating whether a particular demographic likely did or likely did not (with $p < 0.05$) influence the distribution of the result. These results alone do not indicate how the distribution was impacted and the magnitude of the effect. The influence of various demographics are detailed in the results and the implications included in the discussion section.

Using a combination of the survey and statistical results (both primary questions and demographics), and further informed by the exploratory information on SLR priorities and preferences in the pilot study described in *Carpenter (2018)*, six plain-language key findings were constructed (one for each major theme discussed in the survey). These key

findings were developed with the intention of use by public officials, and therefore they summarize key information within the study in an action-oriented manner. The specific justification for each finding is included in the discussion.

## RESULTS

### Survey responses

503 respondents completed the survey including the consent form in question 1 (Q1). 235 respondents reported living in a coastal community, 69 working in a coastal community, and 284 regularly visiting a coastal community (Q2). These add to more than 503 because some respondents had more than one affiliation. There were several significant differences between residents (235) and non-residents (268), as described in the priorities and vulnerability sections.

To provide context on the perceived importance of sea level rise relative to other issues, respondents were asked to state the importance of ten different key issues (Q3), including both environmental issues (such as preparing for sea level rise or protecting the environment) and non-environmental issues (such as reducing taxes or maintaining roads and other transportation infrastructure), on a Likert-type 1-5 response, with 1 meaning "very unimportant" and 5 meaning "very important". For this and all future questions, the definition of every value in the Likert-type ranking, the full question, and each selection is available in the "questions and response meanings" in the supplemental materials (write-in responses are included in the full data set). This question was designed to assess the relative priority of sea level rise planning compared to both issues which are potentially close proxies for sea level rise planning in the public's mind ("preparing for climate change" and "protecting against future flooding") and those that may have some connection but are probably not directly linked in the public's view (such as "helping people with limited resources" and "growing the economy"). Table 2 shows the responses to this question, showing that although "preparing for sea level rise" was one of the lowest ranked issues by both mean (3.68) and percent ranking as 4 or 5 (65.4%), other closely related issues such as "protecting the environment" (4.04 and 75.7%) and "protecting property from natural disasters (3.99 and 75.3%) were ranked more highly.

Survey participants were asked to comment on the importance of various potential components of a local sea level rise plan in a Likert-type 1-5 response (Q4), with 1 meaning "very unimportant" and 5 meaning "very important". This question sought prioritization of sea level rise plan components recognizing that time and resources are likely to be limited. The responses to the question, which are detailed in Table 3, include highly ranked activities around preparing to respond when flooding happens, implementing required policies to mitigate future flood damage, and developing maps and tools to assist in planning. On the other end of the spectrum, fewer than half of participants ranked "finding ways to postpone making change until more research is done" highly, recognizing the need to start planning processes where they have not already begun. This question did not differentiate between postponing all action and developing plans where some decisions can deliberately and constructively be deferred to future under a real options approach (*Dobes, 2008*;

**Table 2** Key issues surveyed sorted by mean score (*Carpenter, 2019*).

| Issue | Mean | Median | Mode | Standard deviation | Number (Percent) ranking 4 or 5 |
|---|---|---|---|---|---|
| Protecting the environment | 4.04 | 4 | 5 | 1.255 | 381 (75.7%) |
| Maintaining roads and other transportation infrastructure | 4.04 | 4 | 5 | 1.220 | 392 (77.9%) |
| Maintaining utilities and related infrastructure | 4.01 | 4 | 5 | 1.200 | 385 (76.5%) |
| Growing the economy | 4.00 | 4 | 5 | 1.198 | 375 (74.5%) |
| Protecting against future flooding | 3.99 | 4 | 5 | 1.248 | 375 (74.5%) |
| Protecting property from natural disasters | 3.99 | 4 | 5 | 1.242 | 379 (75.3%) |
| Helping people with limited resources | 3.90 | 4 | 5 | 1.226 | 368 (73.2%) |
| Reducing taxes | 3.77 | 4 | 5 | 1.255 | 331 (65.8%) |
| Preparing for sea level rise | 3.68 | 4 | 4 | 1.274 | 329 (65.4%) |
| Preparing for climate change | 3.68 | 4 | 5 | 1.302 | 318 (63.2%) |

**Table 3** Sea level rise components surveyed sorted by mean score (*Carpenter, 2019*).

| Component | Mean | Median | Mode | Standard deviation | Number (Percent) ranking 4 or 5 |
|---|---|---|---|---|---|
| Preparing to respond and/or evacuate when flooding happens | 4.11 | 5 | 5 | 1.192 | 392 (77.9%) |
| Implementing required policies to reduce future flood damage | 3.98 | 4 | 5 | 1.171 | 369 (73.4%) |
| Developing maps and tools to learn where flooding will and won't likely cause damage | 3.96 | 4 | 5 | 1.132 | 369 (73.4%) |
| Educating the community on the causes of flooding and sea level rise | 3.88 | 4 | 5 | 1.209 | 355 (70.6%) |
| Building physical barriers (sea walls, levies, dunes, etc.) to protect against flooding | 3.87 | 4 | 5 | 1.247 | 357 (71.0%) |
| Calculating the most cost-effective places and things to protect | 3.85 | 4 | 5 | 1.182 | 350 (69.6%) |
| Working in the community to implement voluntary protections | 3.82 | 4 | 4 | 1.123 | 350 (69.6%) |
| Finding ways to postpone making changes until more research is done | 3.27 | 3 | 3 | 1.262 | 218 (43.3%) |

*Hoekstra & De Kok, 2008*; *Linquiti & Vonortas, 2012*). This phenomenon could be explored in more detail during the development of SLR plans. Closely related to this question was the following one (Q5), which asked participants to write-in any additional SLR plan components, of which the most common response (20.8% of 120 coded responses) was education.

Respondents were asked about their perceived vulnerability to four natural hazards—water surge damage, repeated flooding from high tides, increased flooding from SLR, and other natural disasters (Q6) on a Likert-type question, with 1 meaning "not at all vulnerable" and 5 meaning "exceptionally vulnerable". This question gauged how vulnerable respondents felt about these topics, rather than any objective measure of vulnerability (which would have required data not collected in this study). Figure 1 shows
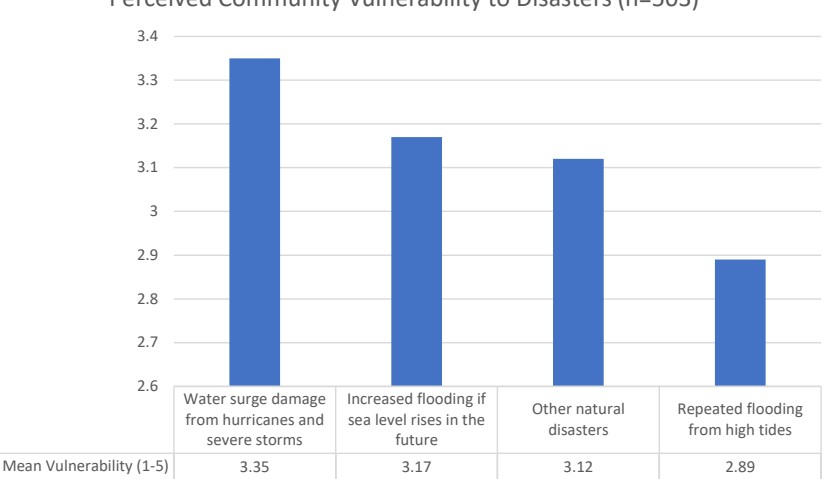

Perceived Community Vulnerability to Disasters (n=503)

| | Water surge damage from hurricanes and severe storms | Increased flooding if sea level rises in the future | Other natural disasters | Repeated flooding from high tides |
|---|---|---|---|---|
| Mean Vulnerability (1-5) | 3.35 | 3.17 | 3.12 | 2.89 |

**Figure 1  Perceived community vulnerability to disasters ($n = 503$).**

mean vulnerability for each component of this question (more detailed information is shown in Table S1). Overall, respondents found themselves to be the most vulnerable on average to damage from hurricanes and severe storms but also perceived some vulnerability from other hazards.

Respondents were presented with 15 potential protection priorities (which services and places to focus protection on with a SLR plan) (Q7). The intent of this question was to help identify what types of local services and amenities are considered by the public to be the highest priorities for inclusion in a sea level rise plan. These included a wide range of options, including various types utilities and related infrastructure (drinking water, electric power, sewer/wastewater and others), as well as individual homes, places of cultural importance, and others. These results are shown in Table 4. Although many essential services and others were highly ranked, drinking water was the only to exceed 80% of respondents ranking highly. Electric power, roads and highways, homes and residences, and sewer/wastewater were all at greater than 70% ranking as 4 or 5. When respondents were asked to identify other priorities not listed (Q8), those with more than five responses include medical facilities/hospitals (11 responses), educational facilities/school (10), and animal shelters/zoos (7). It is important to recognize that many services and amenities are ultimately interdependent within a community and may also rely upon actions within other communities, and therefore a prioritization is only a starting point for more in-depth local assessment.

To help better understand how the public perceives that local sea level rise plans can be developed, funded, and administered, respondents were asked about their preferences on whether the responsibility for preparing for future flooding and sea level rise should be the responsibility entirely of the public sector, entirely of the private sector, or somewhere between the two (Q9). The distribution of these responses is shown in Fig. 2. In this case, over 60% of respondents (303) selected "equal mix of public and private sectors" and of the

**Table 4  Summary statistics for protection priorities (*Carpenter, 2019*).**

| Priority for protection | Mean | Median | Mode | Std. Dev. | Number (Percent) ranking 4 or 5 |
|---|---|---|---|---|---|
| Drinking water | 4.30 | 5 | 5 | 0.994 | 413 (82.1%) |
| Electric power | 4.23 | 5 | 5 | 0.957 | 399 (79.3%) |
| Roads and highways | 4.07 | 4 | 4 | 0.899 | 386 (76.7%) |
| Homes and residences | 4.07 | 4 | 5 | 1.020 | 380 (75.5%) |
| Sewer / wastewater | 3.97 | 4 | 5 | 1.085 | 352 (70.0%) |
| Government facilities | 3.90 | 4 | 5 | 1.042 | 343 (68.2%) |
| Natural gas / heating fuel | 3.85 | 4 | 4 | 1.089 | 337 (67.0%) |
| Beaches and similar coastal amenities | 3.75 | 4 | 4 | 1.120 | 319 (63.4%) |
| Natural wetlands, wildlife areas | 3.71 | 4 | 4 | 1.192 | 318 (63.2%) |
| Stormwater and green infrastructure | 3.69 | 4 | 4 | 1.036 | 313 (62.2%) |
| Businesses, offices, shops | 3.67 | 4 | 4 | 1.059 | 300 (59.6%) |
| Public transit | 3.62 | 4 | 4 | 1.180 | 296 (58.8%) |
| Places of cultural importance | 3.47 | 4 | 3 | 1.076 | 254 (50.5%) |
| Parks and public spaces | 3.43 | 3 | 3 | 1.120 | 241 (47.9%) |
| Houses of worship | 3.31 | 3 | 3 | 1.254 | 234 (46.5%) |

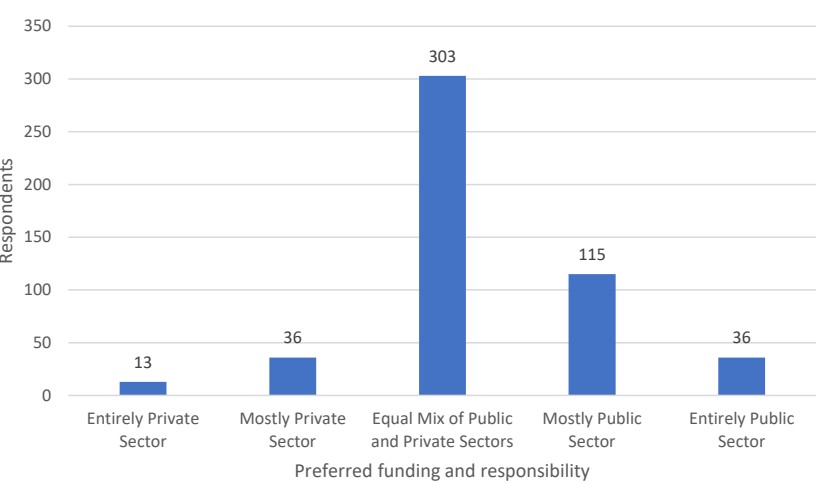

**Figure 2  Preferred funding and responsibility for future flooding and sea level rise ($n = 503$) (*Carpenter, 2019*).**

remaining, more selected "mostly public sector" (115) than any other choice. This suggests that although the public generally looks more towards the public sector, the private sector is also viewed as a key contributor in SLR planning.

Recognizing that funding can be a significant challenge for implementation of any community-wide project, whether or not related to SLR, respondents were asked about the usefulness of various funding mechanisms in their communities (Q10), going from

**Table 5  Summary of responses to funding mechanisms (*Carpenter, 2019*).**

| Funding methodology | Mean | Median | Mode | Standard deviation | Number (Percent) Ranking 4 or 5 |
|---|---|---|---|---|---|
| Hold public meetings to identify highest priorities and vote on methods to pay for them | 3.64 | 4 | 4 | 1.101 | 298 (59.2%) |
| Minimize the use of local taxes but utilize state/federal money when available | 3.56 | 4 | 4 | 1.088 | 275 (54.7%) |
| Encourage insurance companies to require upgrades on homes/businesses to reduce risks as a condition of insurance | 3.41 | 3 | 3 & 4 (Tied) | 1.167 | 248 (49.3%) |
| Set policies to encourage individuals / businesses to pay for their own protection to minimize local government costs | 3.27 | 3 | 4 | 1.211 | 230 (45.7%) |
| Increase funding by raising local fees for beaches and other amenities | 3.05 | 3 | 3 | 1.216 | 189 (37.6%) |
| Use only money already used for protection (no change) | 2.96 | 3 | 3 | 1.297 | 175 (34.8%) |
| Increase funding by raising local sales taxes | 2.83 | 3 | 3 | 1.256 | 161 (32.0%) |
| Increase funding by raising local property taxes | 2.76 | 3 | 2 | 1.290 | 149 (29.6%) |
| Increase funding by raising local income taxes | 2.69 | 3 | 3 | 1.294 | 137 (27.2%) |
| Increase funding for protection by cutting other local programs and services | 2.62 | 3 | 1 | 1.396 | 140 (27.8%) |

"not at all useful" (1) to "exceptionally useful" (5). Recognizing that the actual funding need and availability will vary considerably based upon other aspects of the SLR plan and other conditions within and outside of the community (such as availability of various funding programs), this question focused on the perception of how useful various funding mechanisms are, rather than the actual funding amounts or anticipated availability. Ten options were available, including voting on methods to pay for highest priorities, increasing various forms of taxes, and others. The summary of these responses can be seen in Table 5. Although no funding mechanism had greater than 60% of respondents rank it in one of the top two rankings, "hold public meetings to identify highest priorities and vote on methods to pay for them" was the closest, obtaining 59.2% of responses ranking as 4 or 5. On the opposite end, all forms of increased local taxes received a ranking of 4 or 5 by fewer than one third of respondents. Respondents were also given the opportunity to describe other mechanisms (Q11). Notable amongst these are responses that can be categorized as "governmental action/funding" (6), "improved information" (6), and "donations/fundraising" (6).

Recognizing that there are tradeoffs (including cost, complexity, and level of protection) to all hazard mitigation decisions, respondents were asked to indicate the desired level of protection (which could also be interpreted as tolerance for failure of those protections) from the cumulative protections of their SLR plan. Respondents were asked from ranges as frequent as failing less than 1 in every 10 years all the way up to failing less than one in every 1,000 years, and they were asked both about minor flooding and major flooding. Figure 3 shows the distribution of responses, including a general preference for failure less than 1 in 100 years for both major and minor flooding, although there considerable distribution across the choices, with protections generally desired to be stronger for addressing major

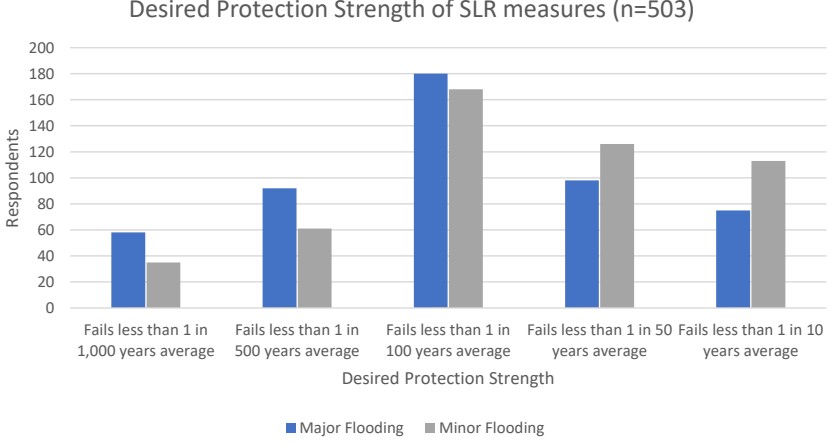

**Figure 3  Desired protection strength of SLR measures (*n* = 503).**

flooding than for minor flooding (more detailed information is available in the Table S2). Although the question noted that protective measures that are less likely to fail are likely to be more expensive and complicated, there was no way to identify exactly how much so for the purposes of this survey. Rather, communities would need to study the specific tradeoffs amongst various protective measures as part of their planning processes.

Some of the decisions that will need to be made in developing and implementing a local sea level rise plan will probably be controversial. For this reason, respondents were asked to rate how helpful eight different methods to resolve conflict locally are likely to be, with 1 meaning "not at all helpful" and 5 being "exceptionally helpful" (Q13). Of the methods, the most favorable was considered to be discussions with preparedness experts about improving protection against floods. Discussions with scientists and increasing educational efforts were also high on the list. Table 6 shows these results. Respondents were also asked to write in any other methods that may be effective for resolving conflict (Q14). The most common responses coded to "community meetings" with eight, over 25% of all of the write-ins for this question.

The last two primary questions were around the perceived appropriateness of various adaptation responses that could be undertaken to increase resilience while adapting to SLR. The first question addressed these from a list (Q15) and the second allowed for additional write-in responses (Q16). In Q15, the responses ranged from "very inappropriate" (1) to "neither appropriate nor inappropriate" (3) and finally to "very appropriate" (5). Of a list of ten adaptation options, including a wide range of choices such as early warning tools, raising elevations on new construction (and/or existing construction), and hardening public infrastructure, all options were generally considered appropriate by having medians above three (neutral), except for "increase cost of insuring high-risk areas" with a median

**Table 6  Summary of methods to resolving conflict by mean score (*Carpenter, 2019*).**

| Conflict resolution methodology | Mean | Median | Mode | Std. Dev. | Number (Percent) ranking 4 or 5 |
|---|---|---|---|---|---|
| Discuss with preparedness experts about ways to improve protection against floods | 3.85 | 4 | 4 | 1.044 | 336 (66.8%) |
| Discuss with scientists about the chances and locations of future flooding | 3.80 | 4 | 5 | 1.107 | 317 (63.0%) |
| Increase educational efforts through the media about the risks and impacts of flooding | 3.80 | 4 | 4 | 1.082 | 324 (64.4%) |
| Start with measures that have the greatest public support | 3.75 | 4 | 4 | 1.044 | 317 (63.0%) |
| Perform cost and benefit analysis on various ways to move forward | 3.70 | 4 | 4 | 1.012 | 303 (60.2%) |
| Hold public meetings to identify ways to resolve conflicts | 3.61 | 4 | 4 | 1.083 | 284 (56.5%) |
| Hold votes on options to resolve disputes | 3.47 | 4 | 4 | 1.132 | 259 (51.5%) |
| Make some measures optional for individual homes and businesses | 3.34 | 3 | 3 | 1.200 | 238 (47.3%) |

**Table 7  Summary of appropriateness of responses to flooding and SLR by mean score (*Carpenter, 2019*).**

| Response for gauging appropriateness | Mean | Median | Mode | Std. Dev. | Number (Percent) ranking 4 or 5 |
|---|---|---|---|---|---|
| Develop and enhance early warning systems to notify residents about upcoming floods | 4.20 | 4 | 5 | 0.943 | 401 (79.7%) |
| Develop and enhance natural physical barriers (such as wetlands or sand dunes) | 4.17 | 4 | 5 | 0.937 | 397 (78.9%) |
| Harden public infrastructure (roads, utilities, etc.) against damage | 4.13 | 4 | 5 | 0.896 | 390 (77.5%) |
| Develop and enhance man-made physical barriers (sea walls, levies, etc.) | 4.07 | 4 | 4 | 0.967 | 393 (78.1%) |
| Require new structures to be built at higher elevations | 4.07 | 4 | 5 | 0.970 | 382 (75.9%) |
| Prevent new development on the most vulnerable areas | 4.00 | 4 | 5 | 1.091 | 360 (71.6%) |
| Raise the elevation of existing structures | 3.73 | 4 | 4 | 1.025 | 308 (61.2%) |
| Remove existing development from the most vulnerable areas over time | 3.50 | 4 | 4 | 1.182 | 271 (53.9%) |
| Increase cost of insuring high-risk areas | 3.42 | 3 | 3 | 1.183 | 247 (49.1%) |
| Don't provide assistance for areas at highest risk | 2.52 | 2 | 1 | 1.419 | 140 (27.8%) |

of 3, and "don't provide assistance for areas at highest risk" with a median of 2. Table 7 shows these responses.

There was limited participation in identifying other potential adaptation measures ($n = 21$ respondents with a total of 26 responses). Of those that did respond, six coded to "improve public infrastructure" which is very similar to one of the responses in the previous question.

Part of the survey was the collection of a series of pieces of demographic information self-reported by each respondent. These included a household income range, self-rated level of environmentalism, job title, level of education, ethnicity, political affiliation, age range (Q24), gender (Q25), location/state (Q26). Summaries of these responses are included in

Tables S3–S10. Respondents were also asked to provide any feedback of concerns about the survey (Q23), for which the most common responses (other than responses indicating no feedback) were either something positive about the survey (17) or concerns about sea level rise itself (11) or the wording of the survey (7).

The primary use of the demographic information was to analyze similarities and differences in priorities across demographics, presented in the statistical analysis below. Therefore, the demographic information is not presented in full here but is available in the supplemental materials. Overall, although some of the distributions are not perfectly representative of the underlying population (for example, female respondents are overrepresented compared to the underlying population), they are diverse enough to represent a substantial number of viewpoints.

## Statistical analysis

There were several key differences between those respondents who are residents of coastal communities and those who are not (i.e., those who either worked in or regularly visited a coastal community but were not also residents). Of the 503 respondents, 235 lived in coastal communities and 268 did not. 65 potential differences (all of the subcategories of each primary question) were tested using Mann–Whitney U (nonparametric) tests. Five relationships which were found to be significant. The following five statements are provided first with the Mann–Whitney U $p$-value.

- Residents perceived their communities to be more vulnerable to hurricanes and severe storms (3.57) than non-residents perceived (3.16) the communities they worked in or regularly visited to be ($p = .001$), with a difference in mean of 0.191 to 0.628 (95% confidence interval).
- Residents perceived their vulnerability to repeated flooding from high tides (3.10) to be greater than non-residents (2.70) perceived the communities they were associated with ($p = .001$), with a difference in mean of 0.161 to 0.631.
- Residents perceived their vulnerability to increased flooding if sea level rises in the future (3.40) as higher than non-residents (2.70) perceived the coastal communities they were affiliated with ($p < .001$), with a difference in mean of 0.217 to 0.667.
- Residents placed higher priority on the importance of electric power for sea level rise plans (4.34) than non-residents (4.13) placed on electric power ($p = .016$), with a difference in mean of 0.046 to 0.381.
- Beaches and similar coastal areas were given a higher priority by residents (3.91) than non-residents (3.62) ($p = .008$), with a difference in mean of 0.095 to 0.486.

Figure 4 shows these significant relationships. As there were five statistically significant differences between coastal residents and non-residents out of 65 relationships (7.7% of relationships), there are important differences between the two groups, especially with regards to perceived vulnerabilities (3 of 5 significant relationships) but the overall difference in priorities and preferences was modest.

Some demographics correlated with changes in the distribution of responses for some or all components of primary questions. Overall, the self-reported level of environmentalism
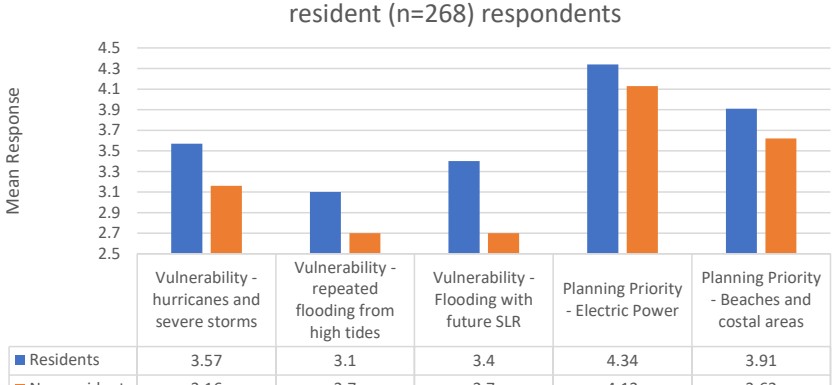

**Figure 4**  **Significant differences between resident ($n = 235$) and non-resident ($n = 268$) respondents.**

**Table 8  Overall influence of each demographic on survey sub-questions by question category (*Carpenter, 2019*).**

| Overall influence by percentage | Vulnerability | Funding | Responses | Conflict resolution | Priorities | Issues | Components | Total |
|---|---|---|---|---|---|---|---|---|
| Environmentalist | 100% | 90% | 50% | 100% | 60% | 20% | 38% | **62%** |
| Funding mixture | 25% | 50% | 70% | 13% | 47% | 0% | 13% | **34%** |
| Gender | 0% | 10% | 60% | 13% | 0% | 50% | 75% | **29%** |
| Age | 100% | 60% | 50% | 13% | 7% | 10% | 13% | **29%** |
| Work coastal | 75% | 40% | 0% | 0% | 20% | 10% | 0% | **17%** |
| State | 75% | 20% | 0% | 13% | 27% | 0% | 0% | **15%** |
| Political party | 0% | 30% | 10% | 13% | 7% | 20% | 0% | **12%** |
| Live coastal | 75% | 0% | 0% | 0% | 13% | 0% | 0% | **8%** |
| Visit coastal | 75% | 0% | 0% | 13% | 7% | 0% | 0% | **8%** |
| Ethnicity | 25% | 10% | 0% | 13% | 0% | 10% | 0% | **6%** |
| Education | 0% | 0% | 0% | 0% | 0% | 30% | 0% | **5%** |
| Income | 0% | 10% | 0% | 0% | 0% | 0% | 0% | **2%** |
| **Total** | **46%** | **27%** | **20%** | **16%** | **16%** | **13%** | **11%** | **19%** |

predicted the largest number of changes to the distribution of primary question responses, with 62% of components to questions likely influenced by this demographic. The preferred funding mixture (public, private, or equal mix) was the second most powerful predictor, coming in at 32% of components to primary questions. Gender and Age were each 29%. Notably, ethnicity, education, and income all influenced less than 10% of question components, and political party only 12%. A summary of these results is shown in Table 8. The component tables showing the *P* value of every test on each component of each question are found in the supplemental materials.

### Key findings summary

Six key findings were developed as described in the methods section. The justification for each finding is discussed within the following discussion section. The key findings were:

Finding 1 on relative priority: "Officials are likely to gain better engagement with the public if they make a strong connection between planning for sea level rise and other high priority issues like the environment, infrastructure/utilities, and the economy". This finding is informed by the results to question 2 and Table 2.

Finding 2 on planning components: "Officials should consider building sea level rise plans that integrate response planning and preparedness with mandatory policies to reduce future damage. Maps and tools, educational resources, and voluntary protections were also popular, but inaction to wait for more research was not popular". This finding is informed by the results to questions 3 and 4 and Table 3.

Finding 3 on protection priorities: "Officials should consider the protection of essential utility and transportation services as some of the highest priorities for protection in sea level rise plans. Residents also rate the protection of individual homes and of government facilities very highly". This finding is informed by the results to questions 7 and 8 and Table 4.

Finding 4 on funding priorities: "Funding may be one of the largest challenges of sea level rise planning. Officials should consider public meetings to discuss how to pay for priorities, should use state and federal funds when available, and should work with the insurance industry on risk reduction measures. Officials should avoid cutting other programs and should proceed cautiously with taxes". This finding is informed by the results to questions 10 and 11 and Table 5.

Finding 5 on conflict resolution: "To help prevent and resolve conflict, officials should consider bringing in both preparedness experts and scientists familiar with flooding and sea level rise to talk with the community and use the media to help educate the community about the issue. Avoid making adaptation measures optional to avoid conflict". This finding is informed by the results to questions 13 and 14 and Table 6.

Finding 6 on adaptation responses: "Public officials should consider a variety of adaptation responses. Early warning systems, natural and artificial barriers, and hardening infrastructure are among the items respondents generally found to be appropriate. Even some potentially controversial adaptation, such as preventing new development in vulnerable areas were generally viewed as appropriate. Officials should avoid cutting off assistance from high risk areas". This finding is informed by the results to questions 15 and 16 and Table 7.

## DISCUSSION

### Key findings discussion

The six key findings of this study were developed based upon the study's findings and were written in plain language to be of maximum utility for public officials, as discussed above.

For relative priority (finding 1), respondents ranked preparation for sea level rise relatively low on the list of other issues (question 2, shown in Table 2), indicating that in
many instances they may not fully engage unless they make connections to other issues that are higher priorities, such as the environment more generally or the economy. The long-term and somewhat abstract nature of sea level rise may put it in the back of people's minds, they may not fully understand it, or other barriers may lead to challenges for public officials in engaging the public. SLR plans that link adaptation measures for SLR with other concerns such as maintaining and building economic opportunities, protecting other facets of the environment, and others may assist in gaining needed engagement. Additional study on this phenomenon could yield additional insights, as the reasoning for respondent's answers is not known from this study alone.

For planning components (finding 2), a wide variety of components, such as response plans, mandatory mitigation policies, and maps and tools were popular for respondents, hence the relatively large number of suggestions for public officials to consider (questions 3 and 4, shown in Table 3). The only option ranked considerably lower than the rest was waiting to take action until additional research is done, which was not a popular choice. Additional work could help to identify the reasoning for this, and to better understand the opportunities for informed, strategic decisions that can be made at a later time (under a real options or another decision framework). How and where to incorporate sea level rise planning into other processes (whether in a stand-alone plan, incorporated into other plans, or through some other means) is likely to be a very local decision, given the wide array of potential items to include.

For protection priorities (finding 3), drinking water and electric power were both ranked with a median and mode of 5 (exceptionally high priority), making utilities key candidates for adaptation measures (questions 7 and 8, shown in Table 4). A number of other services (such as those related to transportation) ranked highly. However, it is also possible that officials will have a difficult time prioritizing certain areas and services over others, as most items polled were identified as high priorities, and many services are likely to be highly interdependent on each other.

For funding priorities (finding 4), the preferred method to identify funding was to hold public meetings (questions 10 and 11, shown in Table 5). Although public meetings may indeed be useful, this finding also poses the challenge that the identified funding sources (for example, using federal and state funds, which also ranked highly) may not be available when and in the quantities desired to meet the needs identified by public engagement. Local methods to raise funding (e.g., taxes) were potentially controversial and may pose challenges in gaining support, despite the greater level of local control.

For conflict resolution (finding 5), both discussions with preparedness experts and scientists were amongst the most popular choices (questions 13 and 14, shown in Table 6). A number of other means to prevent or resolve conflict (such as starting with measures that have the greatest public support, holding public meetings, and others) had similar levels of popularity, meaning that a number of conflict resolution methods may be acceptable to the public in sea level rise planning.

Finally, for adaptation responses (finding 6), most of the surveyed adaptation measures were generally considered acceptable by the respondents (questions 15 and 16, shown in Table 7). This included several measures that were expected to be controversial, such as

preventing new development and removing existing development from vulnerable areas over time, were also generally acceptable. The only clearly unacceptable response of those studied was not providing assistance for areas at highest risk.

### Impact of demographics

As mentioned in the results section, many demographics, such as income, education, and ethnicity had little impact on the distribution of responses for most components of most questions. Rather, the perceived level of environmentalism, preferred (public/private/mixture) funding sources, gender, and age impacted the most components of the questions. Although involving a diverse group of stakeholders across all demographics is essential to full engagement of a community, assuring a solid mixture of individuals affiliated with those demographics that have the greatest influence could be especially important in local sea level rise planning. By assuring, for example, that groups with differing viewpoints on environmental matters are represented in the process, there is greater potential for building buy-in through the process rather than ending in conflict.

## CONCLUSIONS

Developing a locally-driven sea level rise plan is likely to be a challenging process, involving technical expertise, policy tradeoffs, and considerable community input. The six key findings and related information from this study can be used by public officials on the East Coast of the United States and elsewhere to better engage the public on this difficult but necessary process, by better understanding the general priorities and preferences of others affiliated with these coastal communities.

Much additional work can be done to further advance these issues. First, similar studies could be conducted elsewhere the in United States (for example, in Gulf states or on the West Coast) or in nearly any country that has one or more coastal regions. Additional study can help to validate the usefulness of these key findings with policy makers, through discussions or by utilizing them in public processes and evaluating their effectiveness. Although the pathway to coastal resilience through sea level rise planning will likely be difficult, through the development of tools and resources such as this study, public officials can better understand how to get started and some strategies for success.

## ACKNOWLEDGEMENTS

I would like to thank all the members of my dissertation committee who provided valuable guidance and advice during the development and execution of this research: Dr. Dann Sklarew, Dr. Sharon deMonsabert, Dr. Robert Jonas, Dr. Barry Liner, and Dr. Paul Schopf.

### Funding

The author received no funding for this work.

## Competing Interests

Adam T. Carpenter is an employee of the American Water Works Association (Washington, DC, United States).

## Author Contributions

- Adam T. Carpenter conceived and designed the experiments, performed the experiments, analyzed the data, prepared figures and/or tables, authored or reviewed drafts of the paper, and approved the final draft.

## Human Ethics

The following information was supplied relating to ethical approvals (i.e., approving body and any reference numbers):

The George Mason University Institutional Review Board reviewed and approved the collection of this survey data (Approval 1168842-1).

## Data Availability

The survey data supporting this manuscript are available as a Supplemental File and at FigShare: Carpenter, Adam (2019): Public survey on priorities and preferences in developing locally driven sea level rise plans in eastern coastal states in December 2017. figshare. Dataset. https://doi.org/10.6084/m9.figshare.9547298.v1.

## Supplemental Information

Supplemental information for this article can be found online at http://dx.doi.org/10.7717/peerj.9044#supplemental-information.

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
