# Peer review of "Public priorities on locally-driven sea level rise planning on the East Coast of the United States"

_PeerJ, doi:10.7717/peerj.9044_

## Round 0.1 · original submission · Major Revisions

The following are the revisions that I think are most important:
1. Providing more information on how the sample was selected.
2. Drop the parametric tests as suggested by Referee #2
3. Relate your findings to the existing literature on climate change adaptation rather than just as a list of policy prescriptions. This is I think what Referee #2 is getting at when he says the conclusions are atypical.
4. Provide a detailed response to the referees - you need to either make their suggested changes or explain why you are not doing so.

Reviewer 1 ·

Basic reporting

Attached

Experimental design

Attached

Validity of the findings

Attached

Annotated reviews are not available for download in order to protect the identity of reviewers who chose to remain anonymous.

Reviewer 2 ·

Basic reporting

I have now reviewed the manuscript authored by Dr. Carpenter titled: “Public priorities on locally driven sea level rise planning on the East Coast of the United States”. In this manuscript Dr. Carpenter present and analyses survey data obtained from 503 people affiliated with coastal communities on the east coast of the US. The goal of the survey was to assess the values of the public and stakeholders on sea level rise planning. I believe that the information collected and synthesized by Dr. Carpenter could be helpful for management and planning efforts. The English was clear and unambiguous. References appeared suitable, although: you may want to briefly (1-2 sentences) describe likert-type scales and provide citation.

I believe that the author should consider improving the presentation of his information. I would encourage synthesizing the information in figures, which may be a better way to communicate (e.g., findings about vulnerability to hurricanes for residents vs non residents could be plotted in a bar chart to contrast the differences, the table could still be included), pie charts (or other figure types may also help). Providing high quality and informative figures may make the article more appealing to readers. I would suggest that the author refer to tables, figures (or provide effect sizes) when making a statement, such as on lines 268-270 or to justify each finding. I found the discussion and conclusion very atypical, and I would suggest improving these sections. I would also justify each findings by referring to figures, tables and statistics.

Experimental design

This was a basic survey and randomization was not considered for this study
The statistical analyses were also fairly basic (e.g., a more thorough analysis would have possibly involved the use of hierarchical models) but given the descriptive nature of this manuscript, these advanced analyses may not be necessary.

Validity of the findings

With respect to the choices of tests, I would suggest picking one test or the other (parametric or non-parametric but I wonder why include both). I would also provide the appropriate tests for the t-test (homogeneity of variance [the author mentioned Levene’s test but it is not included in the results], and normality test, the two group means and standard deviations. By the way, these tests do not provide information about the magnitude of the effect, so it may be useful to compute the effect size (e.g., arithmetic difference and associated uncertainty, e.g., 95%CI around the estimate of effect size).

Additional comments

Minor Comment:

l. 31 you may want to add measure of uncertainty.
I would suggest improving the format of the figure, in my opinion it does not meet publication standards.
You may want to briefly (1-2 sentences) describe likert-type scales and provide citation.
l-155 to 157, I would think that this information could be moved to the methods section (instead of the results).

---

## Round 0.2 · Minor Revisions

The second reviewer submitted comments to me via email. They are as follows:

"I have re-read the article, but am still left with the overwhelming feeling that its content is largely procedural (albeit necessary in explaining the methods used for the survey) and descriptive. It seems to lack a hard analytical edge, although some readers may find the survey results alone to be valuable.

For example, the author has now taken account of real options by citing a couple of papers (e.g. Linquiti & Vonortas), but this is not tied in with an actual survey finding (lines 470 to 471) about the lowest ranked option being waiting to take action until additional research is done (a real option). Had the analytical work been done first, then the survey could have been designed to elicit reasons as to why people preferred immediate action - perhaps because they lacked context about the personal costs (e.g. higher rates or taxes) of doing so?

Overall, I find that this article is still marginal, except that the survey results may be found to be useful by someone.

if publication is envisaged, typos at lines 71, 115, and 134 should be addressed."

You should also address these comments in your revision.

Reviewer 2 ·

Basic reporting

Although, the authors addressed the comments point by point, I feel like some of the added text was unclear and not well written (see two examples below).
"Likert-type responses are a method to measure, among others, things that are perceptions that cannot be directly measured (Boone, 2012; Carifio, 2008). "

"Given the large sample size and clearly marked scale, or these controversies, the analysis of this data takes a hybrid approach by providing descriptive statistics including means, but the means are primarily used to describe the frequencies of responses."

I strongly encourage the author to work on improving the writing of the added text throughout the paper.

Also please clarify the following:
For example, in a real options approach, adaptation measures that can be exercised in the future as scenarios unfold can be developed while acknowledging the uncertainty associated with them (Dobes, 2018; Linquiti and Vonortas, 2012).
>Definition of "real option approach" is unclear.

In my opinion the format of the figure should be improved. I would suggest adding legends to the vertical axes on the figures (Figs. 3 and 4)

Experimental design

no comment

Validity of the findings

no comment

Additional comments

no comment

---

## Round 0.3 · accepted · Accept

I am happy with the changes you have made to the paper.